# Yield Gap Management under Seawater Intrusion Areas of Indonesia to Improve Rice Productivity and Resilience to Climate Change

**Hasil Sembiring [1,2,\*], Nuning A. Subekti [2], Erythrina [3], Dedi Nugraha [2], Bhakti Priatmojo [2] and Alexander M. Stuart [1]**

[1] International Rice Research Institute—Indonesia Office, Jl. Merdeka No. 147, Bogor 16111, Indonesia; a.stuart@irri.org

[2] Indonesian Center for Food Crops Research and Development, Jl. Merdeka No. 147, Bogor 16111, Indonesia; argosubekti@gmail.com (N.A.S.); edgraenterprise@gmail.com (D.N.); bhakti3priatmojo@gmail.com (B.P.)

[3] Indonesian Center for Agricultural Technology Assessment and Development, Jl. Tentara Pelajar No.10, Bogor 16124, Indonesia; erythrina_58@yahoo.co.id

\* Correspondence: h.sembiring@irri.org

**Abstract:** The purpose of this study was to evaluate (a) the performance of two modern rice varieties (non-tolerant and tolerant for saline soils) under different fertilizer management options, and (b) assess the yield gap and income increase through proper crop and nutrient management at different levels of soil salinity. Experiments were carried out in moderate and high levels of soil salinity in West Java, Indonesia. A split plot design with three replications was used. The main plots included two rice varieties, Inpari-30 Ciherang *sub1* and Inpari-34 (tolerant variety for saline soils), and subplots included eight fertilizer management treatments. Farmer participatory field trials were also established across three levels of soil salinity with four different rice varieties, Sidenuk, Inpari 30, Inpari 34, and Inpari 35, and a fertilizer package consisting of organic and inorganic fertilizers. Under low and moderate soil salinities, Sidenuk and Inpari 30 with recommended practice had higher productivity and economic benefit compared to the saline tolerant rice varieties, Inpari 34 and Inpari 35. However, under high soil salinity, the yields of Inpari 34 and Inpari 35 with recommended practice were 93% higher than farmers' practice, representing an exploitable yield gap of 1.3 t ha$^{-1}$ and benefit above fertilizer cost of USD 301 ha$^{-1}$. The combination of tolerant varieties and improved nutrient management use for rice production can therefore be used as a strategy for improving farmers' income and livelihoods in coastal areas of Indonesia.

**Keywords:** salinity; seawater intrusion; tolerant rice variety; best management practices; yield gap

## 1. Introduction

The Indonesian archipelago comprises 17,504 islands with a total land area of around 1.9 million km$^2$ [1]. Indonesia has a coastline of about 108,000 km$^2$, and about two million people live in coastal areas with an elevation of between 0 m and 2 m above sea level [2]. A substantial proportion of these agricultural areas are within close proximity of the sea [2]. Indonesia's vast coastal agroecological zones are vulnerable to the effects of climate change such as sea level rise [3,4] and require strategies for adaptation, particularly of their rice-based systems [5]. Sustainable improvements in rice production in unfavorable rice ecosystems in the coastal deltas are crucial issues for the Indonesian rice sector, the rural communities, and smallholder farmers [6].

As a consequence of global warming, the frequency and intensity of extreme climatic events, such as flood and drought phenomena, is increasing [3,7]. In addition, an increase in sea level will pose a greater and more frequent flood risk in low-lying coastal areas. With a decrease in the water discharge on the mainland due to climate change leading to reduced rains and increased evaporation, increasing urban and industrial demand and seawater infiltration into canals, streams, and swamps will increase. The over-extraction of groundwater can also result in a lowering of the normal water-bearing stratum levels, leading to the intrusion of seawater. This will have an impact on paddy fields bordering the coast, as the risk of contamination with seawater will become greater [5]. This impact will be felt greater during the dry season when water supply is reduced [8].

On the island of Java, about 29% of rice-growing areas are within 10 km of the sea. The north coastline is a major rice-producing region for the country. This region passes through at least four provinces, namely Banten, West Java, Central Java, and East Java, with a path length around 1316 km (Figure 1). Recently, the productivity of paddy fields in the region has decreased, especially during the dry season. In 2017, around 540,000 ha of rice fields in this region were affected by seawater intrusion, with the dry season producing on average 0.65 t ha$^{-1}$ lower yield than during the wet season [1]. A recent focus group discussion conducted with rice farmers in Karawang, West Java, revealed that low rainfall led to increased salinization through capillary rise and saline water intrusion up rivers and canals, reducing yields in salinity affected fields by 4–5 t ha$^{-1}$ compared non-salinity-affected fields [9].

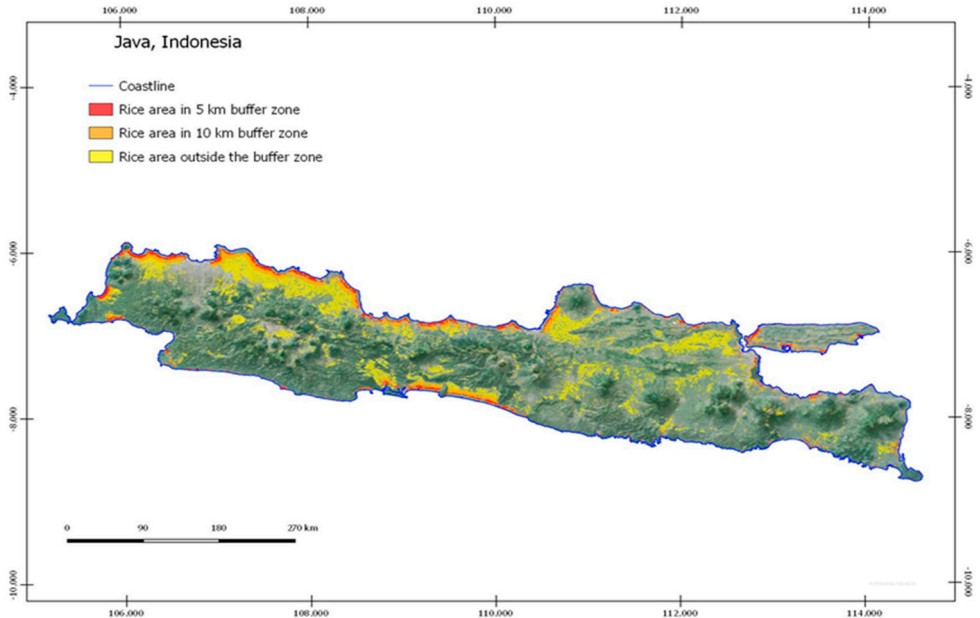

**Figure 1.** Rice growing areas in Java Island with buffer zone within 5 km and 10 km [9].

Research shows that in coastal areas, the soil salinity electrical conductivity (EC) values range from 2 dS m$^{-1}$ to 18 dS m$^{-1}$ during the dry season [10]. The ideal rice tolerance range at planting time is an ECe value of less than 4 dS m$^{-1}$ [11]. Due to poor irrigation supply and saline water intrusion, farmers in some parts of Java have converted their land use from paddy rice into salt-making or fishing ponds or have even abandoned the land [12,13].

There are currently several approaches to alleviate salinity induced problems on rice cultivation in the coastal rice growing area of Java, including the use of soil amendments, chemical fertilizers, organic fertilizers, salt tolerant varieties, and plant growth-promoting bacteria (PGPB) [14–16]. However, clear guidance on an integrated best management approach is currently lacking.

One well-recognized approach is the adoption of salt tolerant rice in salinity prone areas [17]. Traditional breeding approaches to develop salt tolerant varieties have been applied for a long time, but progress has been slow due to the complexity of salt tolerance mechanism and genotype by environment interactions [18]. Through the national rice consortium, two salinity tolerant rice

varieties, Inpari 34 (BR41xIR61920-3B-22-2) and Inpari 35 (IR10206-29-212xSUAKOKO), were released in 2014. These two varieties are tolerant to salinity stress at the seedling stage at an EC value of 12 dS m$^{-1}$ and thus show great potential [17].

Another approach is the use of PGPB. High salt concentration in the soil decreases microbial activity that play a significant role in nutrient cycling [19,20]. The decrease in soil microbial activity in saline soils leads to increased plant stress because of the decreased mineralization rate on nutrients, such as C, N, P, and S, and therefore decreases nutrient availability [21]. PGPB endophytes employ mechanisms similar to those used by rhizospheric plant growth-promoting bacteria. These include both direct and indirect mechanisms, such as nitrogen fixation, ammonia production, solubilization of mineral phosphate, and the production of plant hormones. In addition, plant growth-promoting bacterial endophytes and rhizospheric microorganisms may promote plant growth as a consequence of expressing the enzyme 1-aminocyclopropane-1-carboxylate (ACC) deaminase, which cleaves ACC to a-ketobutyrate and ammonia and thereby decreases ethylene levels in host plants [22,23].

A third approach is the application of gypsum. Gypsum is the most commonly used amendment for saline soil reclamation and for reducing the harmful effects of high sodium irrigation water in agricultural areas because of its solubility, low cost, availability, and ease of handling [24]. The addition of organic matter in conjunction with gypsum has been successful in reducing adverse soil properties associated with saline soils [25,26].

To maintain the sustainability of rice production in Indonesia, it is necessary to re-examine many of the existing approaches to alleviate salinity induced problems in rice cultivation in the coastal rice growing area of Java and develop a set of best management practices. The purpose of this study was to (a) evaluate the performance of two modern rice varieties (non-tolerant and tolerant for saline soils) under different fertilizer management options, and (b) assess the reduction in yield gap and income increase through proper crop and nutrient management at different levels of soil salinity. The best management practices identified in this preliminary study can then be verified and extrapolated to other locations in Indonesia with similar biophysical, climatic, and socioeconomic characteristics.

## 2. Materials and Methods

### 2.1. Description of Study Area

The study was conducted in Indramayu District, West Java Province. Of the approximately 112,000 ha of paddy fields in Indramayu, 16% have high to very high soil salinity, 36% have moderate salinity, and the remaining 48% have low salinity levels [9]. The experiments were conducted at Eretan Kulon village for low salinity level and in Kertawinangun village for moderate and high salinity levels. Both villages are located in Kandanghaur subdistrict, Indramayu. The Kandanghaur subdistrict is geographically located in latitude 6°50′6.72″ and longitude 108°6′15.4794″ and 1.5 m asl.

Fertilizer recommendation for the irrigated rice was developed about two decades ago, mostly as blanket recommendation for single cropping. There is no specific nutrient management for salinity areas [27]. Presently, the cropping system in the region is dominantly rice-rice. Farmers usually plant the rice varieties Inpari 30 (Inpari 30 Ciherang *sub1*) and Sidenuk for both wet and dry seasons. The experiments were conducted during the dry season from August to November 2018.

### 2.2. Soil Sampling and Analysis

A composite soil sample was collected in zigzag manner using an auger to a depth of 20 cm before the start of the experiment. Soil samples were collected before sowing rice in three locations representing different three soil salinity types. One location was in Eretan Kulon Village, which represents low soil salinity, and the other two locations were in Kertawinangun Village, which represent moderate and high soil salinity. Soil samples were analyzed at Indonesian Soil Research Institute Laboratory, IAARD. They were airdried, crushed, and sieved through a 2-mm sieve and

analyzed for their physicochemical properties (Table 1). Based on particle size distribution, soil texture was classified as heavy texture soil with more than 40% clay content. Electrical conductivity (EC) and pH were measured in soil: water extracts of 1:5; CEC was determined by $NH_4$-acetate 1N, pH 7; organic matter was determined by Walkley and Black method; and total N was determined by Kjeldahl method. The value of EC 1:5 for saline land are represented as low, medium, and high, each of which is 0.499 dS m$^{-1}$; 1.728 dS m$^{-1}$, and 2.660 dS m$^{-1}$, respectively. The ECe value was calculated based on the equation:

$$ECe = (14.0 - 0.13 \times \%clay) \times EC1:5 \quad [28] \quad (1)$$

**Table 1.** The ECe value for saline land is 2.02 dS m$^{-1}$, 7.23 dS m$^{-1}$, and 10.51 dS m$^{-1}$ classified as slightly, moderately, and strongly saline, respectively [11]. Physical and chemical properties of studied soil, Indramayu, West Java, DS 2018.

| Property | Soil Salinity | | |
|---|---|---|---|
| | Low | Medium | High |
| Particle size distribution (%): | | | |
| • Clay | 76.6 | 75.5 | 77.3 |
| • Silt | 23.2 | 21.8 | 21.9 |
| • Sand | 0.2 | 0.7 | 0.8 |
| Organic matter (g kg$^{-1}$) | 2.45 | 2.27 | 1.04 |
| Total-N (%) | 0.16 | 0.15 | 0.80 |
| C/N ratio | 15 | 15 | 13 |
| P total (%) | 0.03 | 0.06 | 0.03 |
| K total (%) | 0.38 | 0.43 | 0.55 |
| EC and pH: | | | |
| • EC (dSm$^{-1}$) (Soil paste extract 1:5) | 0.499 | 1.728 | 2.660 |
| • ECe (dSm$^{-1}$) | 2.02 | 7.23 | 10.51 |
| • pH (Soil suspension 1:5) | 6.4 | 6.0 | 6.9 |
| Exchangeable cations, CEC and ESP * | | | |
| • Na$^+$ (cmol kg$^{-1}$) | 2.79 | 7.31 | 13.43 |
| • K$^+$ (cmol kg$^{-1}$) | 0.80 | 1.54 | 2.76 |
| • Ca$^{2+}$ (cmol kg$^{-1}$) | 15.94 | 12.11 | 7.11 |
| • Mg$^{2+}$ (cmol kg$^{-1}$) | 11.86 | 15.00 | 14.07 |
| • CEC (cmol kg$^{-1}$) | 21.80 | 25.63 | 21.70 |
| • ESP. (%) | 13.43 | 21.70 | 61.89 |

* ESP; Exchangeable Sodium Percentage is the relative amount of the sodium ion present on the soil surface, expressed as a percentage of the total Cation Exchange Capacity (CEC).

## 2.3. Field Experiments

### 2.3.1. Best Management Practices (BMP)

Based on justification that BMP recommendations for low-level salinity soil are similar to that of non-saline soils, the experiment was laid out only in two farmers' fields. One field had high saline conditions and the other field had moderate saline conditions. In each farmers' field, a randomized split-plot paired design was applied, with three replications per treatment. Two rice varieties were used as the main plots in this study: (V1) Inpari-30 Ciherang *sub1* and (V2) Inpari-34. Inpari 30 Ciherang *sub1* is a high-yielding rice variety used by farmers while Inpari-34 is a variety with tolerance for saline soil conditions [17]. The subplots were: (1) Farmer fertilizer practices (FFP); (2) Recommended fertilizer package (RFP); (3) BMP1 = RFP + rhizospheric PGRB; (4) BMP2 = RFP + PGRB endophytes; (5) BMP1 without gypsum; (6) BMP2 without gypsum; (7) BMP1 without manure; and (8) BMP2 without manure. The recommended fertilizer package comprised inorganic fertilizers of 100 kg ha$^{-1}$ of NPK (15:15:15), 100 kg ha$^{-1}$ of triple superphosphate (36% $P_2O_5$), 100 kg ha$^{-1}$ of ammonium sulfate (21% N, 24% S), 50 kg ha$^{-1}$ of gypsum (23.3% Ca, 17% S), plus 1 t ha$^{-1}$ of

farm yard manure (0.01% N, 0.01% $P_2O_5$ N and 0.01% $K_2O$). Total fertilizer application was equivalent to 135.5 kg N ha$^{-1}$, 52 kg $P_2O_5$ ha$^{-1}$, 16 kg $K_2O$ ha$^{-1}$, 33 kg S ha$^{-1}$, and 11.5 kg Ca ha$^{-1}$. Farmer fertilizer practices were urea, NPK 15:15:15, and SP-36 (Triple superphospate which contents 36% $P_2O_5$) equivalent to 135 kg N ha$^{-1}$, 76.5 kg $P_2O_5$ ha$^{-1}$, and 22.5 kg $K_2O$ ha$^{-1}$. Farmyard manure contained 0.01% N, 0.01% $P_2O_5$ N, and 0.01% $K_2O$.

Each treatment within a subplot size of 5 m × 6 m was conducted under medium- and high-level soil salinity. The standard plant-population density was 25 cm × 25 cm hill spacing with 2–3 seedlings/hill$^{-1}$ and 30-day-old seedlings as farmers' general practice. Depending on the treatment, gypsum and organic amendments were applied seven days before rice transplanting by incorporating them into the soil. All inorganic fertilizers were applied seven days after planting. Two-split urea-N applications were applied at 24 days and 35 days after planting. Rhizospheric PGRB was obtained from the microbial laboratory of the Indonesian Soil Research Institute-Bogor [29] and PGRB endophytes were obtained from Pajajaran University–Bandung [30,31]. Rhizospheric PGRB inoculation was done by mixing pregerminated seed with microbial culture, followed by drying in shade before planting in the nursery bed. PGRB endophytes were inoculated as a soil application in the nursery bed.

Plant biomass at physiological maturity (about five days before harvesting) was conducted by sampling all rice plants from a 0.5 m$^2$ quadrat and dried until reaching a constant weight at 60 °C. For yield components, rice plants were sampled from a 0.5 m$^2$ area at physiological maturity, and the number of panicles m$^{-2}$, filled and unfilled spikelets, 1000 grain weight, and grain moisture content were recorded using Crown Moisture Meter TA-5. The grain yield was measured from one 9 m$^2$ (3 m × 3 m) sampling area per subplot at harvest. Samples were oven-dried, moisture readings were taken, and crop cut weights were converted to tons per hectare at 14% of moisture content.

### 2.3.2. Yield Gap between BMP and Farmer's Practice

The exploitable yield gap of a crop grown in a certain location and cropping system is defined as the difference between the yield under optimum management and the average yield achieved by farmers [32]. The exploitable yield gap is described as a percentage by dividing this value by the yield under optimum management. The farmer participatory demonstration sites (across three levels of soil salinity) included 66 farmers across a total area of approximately 23 hectares. Each farmer was asked to plant four different varieties, such as Sidenuk, Inpari 30, Inpari 34, and Inpari 35, with one variety planted in each of their natural rice field plots. The area of each plot ranged from 150 m$^2$ to 300 m$^2$. Each cooperative farmer was given free rice seeds and the recommended fertilizer package, consisting of inorganic fertilizers of NPK, triple superphosphate, ammonium sulfate, gypsum, and farmyard manure as organic fertilizer according to the area of rice fields for the demonstration plots. As a comparison, the farmers planted Sidenuk variety in their remaining land and followed their own fertilizer management methods (i.e., Farmer's Practice). Grain yield was recorded from one sampling area of 3 m × 3 m method per plot and was converted to tons per hectare at 14% of moisture content.

The partial budgets were constructed for farmers' current practice and recommended fertilizer package for each of the four rice varieties. Inpari 30 and Sidenuk are high-yielding rice varieties, while Inpari 34 and Inpari 35 are tolerant varieties for saline soils. The purpose of partial budget analysis was to evaluate the differences in costs and benefits among different management systems under low, medium, and high soil salinities. In the preparation of partial budget analysis, not all the costs of production were considered. Instead, only the costs that varied among management practices systems were taken into account. Data were statistically analyzed using analysis of variance, and Duncan's multiple-range test was applied to examine significance of differences between the treatment means. Statistical analysis was conducted using STAR [33].

## 3. Results and Discussion

### 3.1. Best Management Practices (BMP)

### 3.1.1. Grain Yield

Under moderate soil salinity, the grain yield of Inpari 30 was significantly higher than Inpari 34, with mean grain yields exceeding 6 t ha$^{-1}$ for some Inpari 30 treatments (Table 2; Figure 2), whereas the mean grain yields for Inpari 34 per treatment never exceeded 4.5 t ha$^{-1}$. Inpari 30 was selected for improvement from "mega variety" Ciherang (IR18349-53-1-3-1-3/IR19661-131-3-1//IR19661-131-3-1-///IR64////IR64), a widely grown Indonesian cultivar that was developed from multiple variety crosses, including IR64. Inpari 30 is an upgraded version of this variety carrying the sub-1 QTL, namely Inpari 30 Ciherang sub-1 [34,35]. It has been suggested that Inpari 30 mainly uses a tissue tolerance mechanism in response to salt stress [36]. Our findings indicated a clear indication that Inpari 30 has a broad adaptation ability, even under medium soil salinity. However, under high soil salinity, Inpari 34 was superior to Inpari 30, with mean grain yields just exceeding 4 t ha$^{-1}$ for some Inpari 34 treatments (Table 2; Figure 3). Meanwhile, the mean grain yields for Inpari 30 per treatment fell below 3 t ha$^{-1}$ under high soil salinity. On average, the grain yield under high soil salinity for Inpari 34 was around 59% higher than Inpari 30. Barren spots and stunted plants appeared in Inpari 30 growing on high saline areas. The extent and frequency of bare spots is often an indication of the concentration of salts in the soil [37].

**Table 2.** Analysis of variance of different traits under moderate and high soil salinity, Indramayu, West Java, DS 2018.

| Source of Variance | df | Mean Squares | | | | | |
|---|---|---|---|---|---|---|---|
| | | Grain Yield (t ha$^{-1}$) | Above Ground Biomass (t ha$^{-1}$) | Panicle Number | Number of Seed per Panicle | Empty Grain (%) | Weight of 1000 Grains (g) |
| Moderate soil salinity | | | | | | | |
| Variety (A) | 1 | 19,291,980 ** | 19,761,536 ns | 91.8533 * | 7.2385 ns | 1.8252 ns | 0.0675 ns |
| Error (a) | 2 | 224,827 | 13,349,184 | 2.4788 | 202.7430 | 2.0097 | 1.2699 |
| Fertilizer management (B) | 7 | 1,681,251 *** | 14,773,687 ns | 19.3286 *** | 407.7835 *** | 25.4171 * | 2.7255 ns |
| A × B | 7 | 615,247 ** | 548,589 ns | 1.9135 ns | 270.3462 ** | 1.6978 ns | 0.0963 ns |
| Error (b) | 28 | 113,340 | 8,272,480 | 1.8741 | 48.5038 | 10.2057 | 1.2356 |
| cv (a) (%) | | 9.99 | 34.65 | 13.09 | 13.53 | 12.01 | 4.45 |
| cv (b) (%) | | 7.10 | 27.28 | 11.38 | 6.62 | 27.07 | 4.39 |
| High soil salinity | | | | | | | |
| Variety (A) | 1 | 18,585,363 * | 4,404,893 ns | 24.9697 * | 1427.9008 ns | 1623.8970 * | 0.0050 ns |
| Error (a) | 2 | 498,658 | 442,955 | 0.3047 | 141.4758 | 28.8277 | 2.0288 |
| Fertilizer management (B) | 7 | 1,364,059 ** | 6,715,193 * | 11.8847 *** | 494.0164 ** | 33.3816 * | 1.3165 ns |
| A × B | 7 | 135,609 ns | 193,799 ns | 0.5205 ns | 103.9639 ns | 4.9969 ns | 1.1057 ns |
| Error (b) | 28 | 209,854 | 539,870 | 0.5952 | 122.6405 | 11.1116 | 1.6614 |
| cv (a) (%) | | 25.90 | 7.69 | 6.84 | 13.29 | 27.96 | 6.78 |
| cv (b) (%) | | 16.80 | 8.49 | 9.55 | 12.37 | 17.36 | 6.14 |

*, **, *** = Significant at the $p < 0.05$, $p < 0.01$, and $p < 0.001$ probability levels, respectively, ns = non-significant.

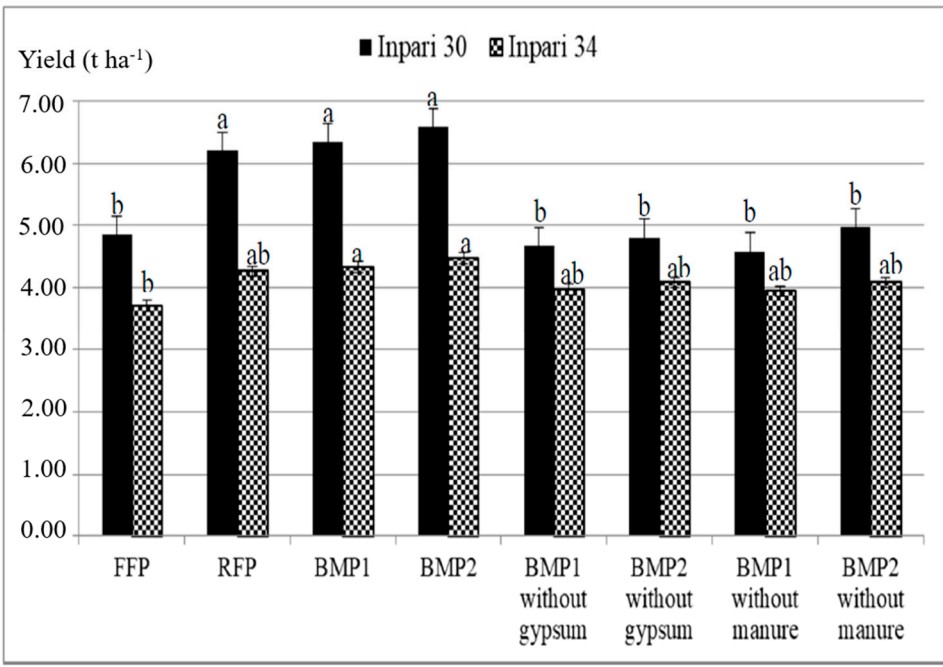

**Figure 2.** Grain yield of rice varieties Inpari 30 and Inpari 34 with different fertilizer management under moderate soil salinity, Indramayu, West Java, DS 2018. For each variety considered, the values followed by the same letter are not significantly different, according to the Duncan Multiple Range Test (DMRT) at $p \leq 0.05$. Error bars indicate 1 standard error of the mean.

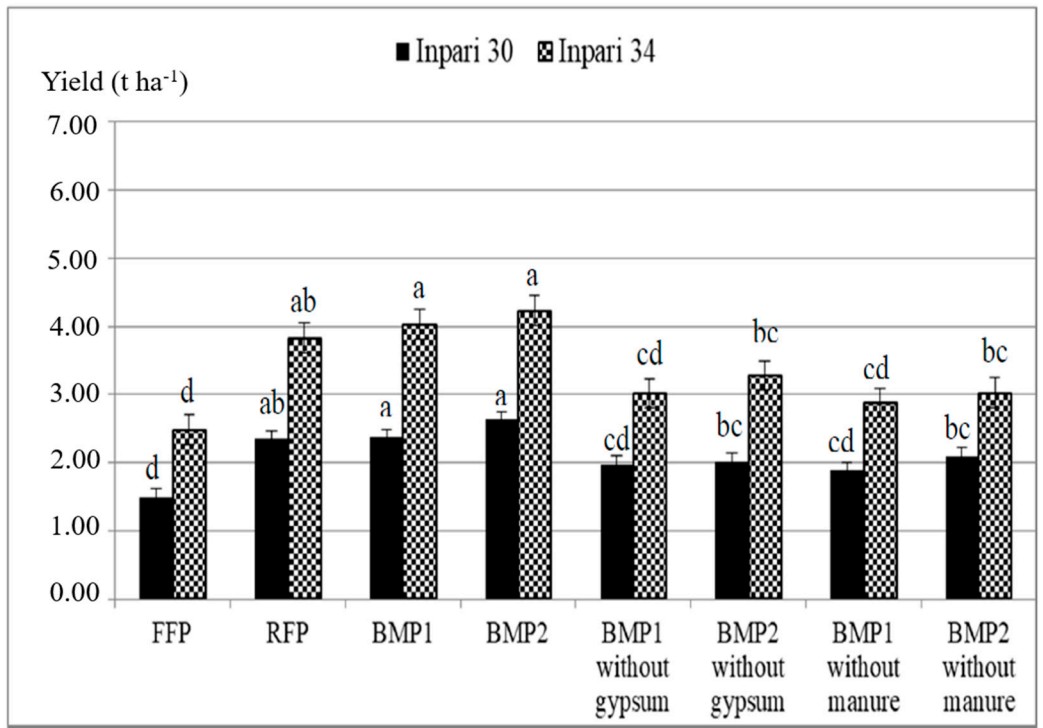

**Figure 3.** Yield of rice varieties Inpari 30 and Inpari 34 with different fertilizer management under high soil salinity, Indramayu, West Java, DS 2018. For each variety considered, the values followed by the same letter are not significantly different, according to the DMRT at $p \leq 0.05$. Error bars indicate 1 standard error of the mean.

Fertilizer management practice had a significant effect on the grain yield for both moderate and high salinity (Table 2). The yields for farmer fertilizer practices (FFP) were the lowest. Under moderate soil salinity, improved nutrient management as recommended fertilizer package (RFP)

with changes of N sources from urea to ammonium sulfate and addition of gypsum and organic fertilizer increased the yield of both varieties Inpari 30 and Inpari 34 by 28% and 15%, respectively. In addition, due to the low yields achieved for BMP 1 and 2 without gypsum and for BMP 1 without manure, there was very little difference in the mean grain yield between the two varieties as indicated by the significant interaction between variety and fertilizer management for grain yield under medium soil salinity (Table 2).

The low productivity of saline soils can be attributed not only to their toxicity due to the salt or to the damage caused by excessive amounts of soluble salts, but also to low soil fertility. The fertility problems are usually evidenced by a lack of organic matter and of available mineral nutrients, especially N and P [30,31]. These soils are also usually characterized by a reduction in the activities of some key soil enzymes, such as urease and phosphatase [38], which are associated with biological transformations and the bioavailability of N and P.

The addition of PGRB rhizozpheric or PGRB endophytes to RFP did not increase the grain yields significantly but showed positive effects. Saline soils are characterized by low organic matter content and reduced organic matter turnover due to poor plant growth and low microbial biomass and activity [19,39]. The availability of nutrients for plants is regulated by the rhizospheric microbial activity. Thus, any factor affecting this community and its functions influences the availability of nutrients and growth of the plants [39].

There was a significant reduction in the grain yield of rice variety Inpari 30 and Inpari 34 without application of gypsum and organic fertilizer compared to best management practices (BMP). Gypsum application was assessed similarly to organic fertilizer treatment in statistical terms. The better ameliorative response of gypsum may be attributed to its rich calcium content, which replaced exchangeable sodium from the soil exchange complex. The replaced sodium leached down as sodium sulphate in the excessive water during land preparation and the rice growing period [40]. Heavy textured soils, as shown in Table 1, and soils with a favorable infiltration rate are likely to respond to gypsum application [11].

The relatively high response of rice crop to organic fertilizer may be attributed to its faster decomposition over time. In saline soils, the organic matter (OM) content is low due to poor plant growth as a result of osmotic stress and ion toxicity. Low input of OM in soils restricts microbial growth by reducing substrate availability [19]. The addition of OM to saline soils can rehabilitate saline soils by improving soil structure, decreasing soil bulk density, and providing energy and nutrients for soil microorganisms [41]. The application of organic matter can accelerate the leaching of NaCl, decrease the percentage of exchangeable sodium and the electrical conductivity, and increase water filtration, the water holding capacity and aggregate stability [42].

In soils affected by salts showing low productivity, the adoption of adequate agricultural practices is of fundamental importance for the success of their exploitation, including modifications in the organic fertilization [43]. The application of decomposing cow manure, straw, or stable manure significantly increased the productivities of rice and wheat cultivated in saline soils [44]. However, the excessive use of organic manure should be avoided, especially in areas flooded for long periods, in order to reduce the risk of toxic effects from reduced intermediates, which accumulate from the anaerobic decomposition of organic manure [45].

### 3.1.2. Yield Components

Excluding 1000-grain weight, the different fertilizer managements had a significant effect on the yield components of rice varieties Inpari 30 and Inpari 34 under both moderate and high soil salinity (Table 2). FFP performed the worst under both soil conditions, with the lowest number of panicles per plant and number of seeds per panicle, and the highest number of empty grains (Table 3). Meanwhile, FFP, BMP1, and BMP2 performed the best. In addition, under high soil salinity, the panicle number, number of seeds per panicle, and weight of 1000 grains were lower compared to plants under moderate soil salinity, while the number of empty grains increased (Table 3). This finding is supported by previous research showing a large influence of soil salinity on yield components [15,46].

**Table 3.** Yield components and above ground biomass of varieties mean at different fertilizer management under moderate and high soil salinity, Indramayu, West Java, DS 2018.

| Fertilizer Management | Panicle Number | Number of Seed per Panicle | Empty Grain (%) | Weight of 1000 Grain (g) | Above Ground Biomass (t ha⁻¹) |
|---|---|---|---|---|---|
| Moderate soil salinity | | | | | |
| Farmer fertilizer practices (FFP) | 9.6 c | 91.3 d | 14.9 a | 24.2 a | 8.849 a |
| Recommended Fertilizer package (RFP) | 13.7 a | 112.7 a | 10.1 bcd | 24.9 a | 11.559 a |
| BMP1 = RFP + PGRB rhizozpheric | 14.1 a | 110.2 a | 9.5 cd | 25.8 a | 12.339 a |
| BMP2 = RFP + PGRB endophytes | 14.3 a | 113.9 a | 9.1 d | 26.5 a | 13.116 a |
| BMP1 without gypsum | 10.1 c | 96.7 c | 13.0 abc | 25.5 a | 9.792 a |
| BMP2 without gypsum | 11.9 b | 103.8 b | 11.9 abcd | 25.0 a | 9.651 a |
| BMP1 without manure | 11.0 bc | 101.5 b | 13.4 ab | 25.4 a | 9.374 a |
| BMP2 without manure | 11.7 b | 101.8 b | 12.5 abcd | 25.5 a | 9.668 a |
| Average | 12.1 | 104.0 | 11.8 | 25.4 | 10.544 |
| High soil salinity | | | | | |
| Farmer fertilizer practices (FFP) | 6.3 d | 82.0 c | 23.5 a | 20.5 a | 6.587 c |
| Recommended Fertilizer package (RFP) | 9.1 b | 95.8 ab | 18.0 ab | 20.5 a | 9.600 ab |
| BMP1 = RFP + PGRB rhizozpheric | 9.4 b | 101.8 a | 16.4 b | 21.3 a | 9.459 ab |
| BMP2 = RFP + PGRB endophytes | 10.4 a | 101.9 a | 16.5 b | 21.6 a | 9.924 a |
| BMP1 without gypsum | 7.4 c | 80.3 c | 20.8 ab | 21.6 a | 8.264 b |
| BMP2 without gypsum | 7.8 c | 83.5 bc | 19.7 ab | 20.8 a | 8.789 ab |
| BMP1 without manure | 7.1 c | 82.2 c | 20.0 ab | 21.0 a | 8.247 b |
| BMP2 without manure | 7.3 c | 88.5 bc | 18.8 ab | 20.8 a | 8.384 b |
| Average | 8.1 | 89.5 | 19.2 | 21.0 | 8.7 |

In a column, means followed by the same letter are not significantly different at the $p < 0.05$.

### 3.1.3. Biomass

Under both soil conditions, FFP had the lowest plant biomass, whereas RFP, BMP1, and BMP2 had the highest (Table 3). However, this difference was only significant under high soil salinity (Table 2). A higher biomass was obtained from moderate soil salinity compared to high soil salinity, representing significant biomass losses due to saline conditions. Radanielson et al. [47] described a variability of responses to salinity in biomass production processes (namely transpiration and photosynthesis) among different rice varieties. However, modified versions of the crop growth models ORYZA v3 and APSIM-Oryza demonstrated an acceptable ability to represent rice biomass and yield production under salt-affected soil conditions [48]. The results of our field experiments can thus be used with either of these models for varietal selection, optimizing crop scheduling, irrigation, and agronomic management, as well as identifying adaptive crop management strategies for rice production in salt-affected areas.

### 3.2. Yield Gap between BMP and Farmer's Practice

Agro-economic analysis based on the average yield of each treatment across 18 farmers' field demonstration plots under low, medium, and high soil salinity revealed yield gaps between the farmers' variety (Sidenuk) with farmers' current practice and the following four rice varieties; Sidenuk, Inpari 30, Inpari 34 and Inpari 35, using recommended practice (Table 4). Under low soil salinity, the average yield using farmers' variety with farmers' current practice was 4.9 t ha⁻¹ compared with 6.1 t ha⁻¹ and 6.3 t ha⁻¹ for Sidenuk and Inpari 30, respectively, with recommended

practice. This represents an exploitable yield gap of 26%. No further yield advantage was observed for tolerant varieties (Inpari 34 and Inpari 35) under low soil salinity. Furthermore, there was a 26% increase in gross margin after deducting fertilizer costs when using recommended practices with farmers' varieties.

**Table 4.** Yield gap and profit analysis between farmer's current practice and recommended practice under low, moderate and high levels of soil salinity, Indramayu, West Java, dry season 2018.

| Item | Low Soil Salinity | | Moderate Soil Salinity | | High Soil Salinity | |
|---|---|---|---|---|---|---|
| | Farmer's Current Practice (*n* =19) | Recom-Mended Practice (*n* = 4) | Farmer's Current Practice (*n* =18) | Recom-Mended Practice (*n* = 4) | Farmer's Current Practice (*n* =17) | Recom-Mended Practice (*n* = 4) |
| Mean Grain yield at 14% m.c. (t ha$^{-1}$) | | | | | | |
| ▪ Sidenuk | 4.937 | 6.137 | 4.075 | 5.350 | 1.294 | 1.489 |
| ▪ Inpari 30 | | 6.299 | | 4.521 | | 1.627 |
| ▪ Inpari 34 | | 5.490 | | 3.953 | | 2.344 |
| ▪ Inpari 35 | | 5.386 | | 3.887 | | 2.638 |
| Yield gap (t ha$^{-1}$) [a] | | 1.362 | | 1.275 | | 1.344 |
| Revenue (USD ha$^{-1}$) [b] | | | | | | |
| ▪ Sidenuk | 2252.04 | 2799.32 | 1830.18 | 2402.81 | 581.16 | 668.82 |
| ▪ Inpari 30 | | 2873.23 | | 2030.48 | | 730.72 |
| ▪ Inpari 34 | | 2504.08 | | 1775.38 | | 1052.72 |
| ▪ Inpari 35 | | 2456.57 | | 1745.74 | | 1184.56 |
| Mean Fertilizer cost (inorganic sources)/ha | 76.84 | 68.77 | 76.84 | 68.77 | 76.84 | 68.77 |
| Fertilizer cost (organic sources) | 0.00 | 35.09 | 0.00 | 35.09 | 0.00 | 35.09 |
| Total cost (USD ha$^{-1}$) | 76.84 | 103.89 | 76.84 | 103.89 | 76.84 | 103.89 |
| Expected benefit above fertilizer costs (USD ha$^{-1}$) | | | | | | |
| ▪ Sidenuk | 2175.20 | 2695.46 | 1753.33 | 2298.95 | 504.32 | 564.96 |
| ▪ Inpari 30 | | 2769.37 | | 1926.62 | | 626.86 |
| ▪ Inpari 34 | | 2400.22 | | 1671.52 | | 948.86 |
| ▪ Inpari 35 | | 2352.71 | | 1641.88 | | 1080.70 |
| Change in benefit (USD ha$^{-1}$) | | | | | | |
| ▪ Sidenuk | 520.26 (23.9%) | | 545.61 (31.1%) | | 60.64 (12.0%) | |
| ▪ Inpari 30 | 594.17 (27.3%) | | 173.29 (9.9%) | | 122.54 (24.3%) | |
| ▪ Inpari 34 | 225.02 (10.3%) | | −81.81 (−4.7%) | | 444.53 (88.1%) | |
| ▪ Inpari 35 | 177.51 (8.2%) | | −111.45 (−6.4%) | | 576.38 (102.3%) | |
| Average | 379.24 | | 131.41 | | 301.02 | |

[a] Yield gap = mean grain yield of highest yielding variety−mean grain yield of farmer's current practice. [b] Based on farm gate price of 0.45 USD kg$^{-1}$; USD = Rp. 14,250.

Similarly, under moderate soil salinity, there was a 28% exploitable yield gap between farmers' variety with farmers' current practice and farmers' varieties grown with recommended practice. The increase in gross margin above fertilizer cost was 21% when using farmers' varieties. However, there was no improvement in yield for tolerant varieties (Inpari 34 and Inpari 35) with recommended practice versus farmers' practice. These results indicate that under low and moderate soil salinities, the modern rice varieties Sidenuk and Inpari 30 performed better than tolerant rice varieties Inpari 34 and Inpari 35 when using recommended practices.

Under high soil salinity, the difference in yield between farmers' varieties with farmers' current practice and Sidenuk and Inpari 30 with recommended practice was only 0.195 t ha$^{-1}$ and 0.333 t ha$^{-1}$, respectively, showing a yield advantage of 20% when using recommended practice. However, when

farmers grew tolerant varieties (Inpari 34 and Inpari 35) with recommended practice, the increase in yield above farmers' practice was 93%, representing an exploitable yield gap of 51%. The increase in gross margin above fertilizer cost was 18% and 87% when using farmers' varieties and tolerant varieties, respectively. Under high soil salinity, the implementation of improved technologies—through either the use of saline tolerant varieties or recommended management—enhances rice productivity, but maximum yield gains could be ensured from combining improved varieties with improved management options. The combination of tolerant varieties and improved nutrient management for rice production can therefore be used as a strategy for improving farmers' income and livelihoods in saline-affected double-rice cropping areas during the dry season.

The use of tolerant rice varieties to remediate saline soils is a low-cost and emergent method. Salt-tolerant varieties are normally more responsive to amendments and mitigation options, and a lack of proper management is often reflected in a yield reduction. Developing rice varieties with wider adaptation and broader tolerance of prevailing stresses is more viable for areas where abiotic stresses are particularly variable and complex and growing conditions are too risky to persuade farmers to invest in inputs.

## 4. Conclusions

Soil salinity is widely reported as the main agricultural problem, particularly in the double-rice cropping of irrigated rice. The use of high-yielding rice varieties such as Sidenuk and Inpari 30 during the dry season under low and moderate soil salinities with recommended fertilizer practice produced the best yields compared to saline tolerant rice varieties. However, under high soil salinity, saline tolerant rice varieties Inpari 34 and Inpari 35 performed better than non-saline tolerant varieties. On average, the exploitable yield gap between farmer's current practice and fields with proper crop and nutrient management (including the application of gypsum and manure) ranged from 1.3 t ha$^{-1}$ to 1.4 t ha$^{-1}$. The increase in benefit after deducting fertilizer cost was USD 379 ha$^{-1}$, USD 131 ha$^{-1}$, and USD 301 ha$^{-1}$ under low, moderate, and high levels of soil salinity, respectively. Overall, there is significant potential for farmers in the saline-affected double rice cropping areas in Indramayu, West Java province, and other provinces across Indonesia with similar biophysical, climatic, and socioeconomic characteristics to increase rice yields by adopting these rice varieties and best management practices. However, this is a preliminary study, and the results need to be verified with additional studies. The study outlays potential for better management with improved stress-tolerant variety in reducing yield gap and increasing income in salinity-prone areas of Indonesia.

**Author Contributions:** Authors as main contributed to the study and manuscript writing H.S., N.A.S., and E. (conceptualization, methodology, and original draft preparation); data collection and analysis, D.N. and B.P.; review and editing, A.M.S. All authors have read and agreed to the published version of the manuscript.

**Funding:** We are grateful for financial support from the Indonesian Agency for Agricultural Research and Development (IAARD) and the World Bank for the Sustainable Management of Agricultural Research and Technology Disseminations (SMARTD) program. The contribution by A.M.S. was financially supported by funding provided by the Swiss Agency for Development and Cooperation for the CORIGAP project (Grant no. 81016734) and by the Australian Centre for International Research (ACIAR) Small Research Activity 'Assessment of management in key coastal areas of Indonesia to improve agricultural productivity and resilience to climate change'.

**Acknowledgements:** We sincerely thank Dr Sheetal Sharma, Soil Scientist at the International Rice Research Institute, and the anonymous reviewers for their helpful comments to improve the manuscript.

**Conflicts of Interest:** The authors declare no conflict of interest. All authors read and approved the final manuscript.

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
