# Peer review of "Yield Gap Management under Seawater Intrusion Areas of Indonesia to Improve Rice Productivity and Resilience to Climate Change"

_agriculture, doi:10.3390/agriculture10010001_

Round 1

Reviewer 1 Report

Dear Authors,

I revised the manuscript "Yield Gap Management under Seawater Intrusion Areas of Indonesia to Improve Rice Productivity and Resilience to Climate Change" submitted to the Agriculture Journal. The paper is very interesting. However, I have some concerns, which need to be addressed.

Line 33. Delete the symbol "-" in "km-2"

Line 116-117. Why did You collect soil samples in only one location at Eretan Kulon and two locations at Kertawinangun? It isn’t clear. Please explain it.

Line 124. Format equation as required in the file "agriculture-template.dot" in section 3.3. Formatting of Mathematical Components.

Line 128. Use number of equation.

Section "3. Results and Discussion". It will be better if You divide this section into two separate sections "Results" and "Discussion".

Line 340-341. "This study, however, is a preliminary study and the results need to be verified with additional studies." Use a similar expression at the end of the section "1. Introduction". It is important to point out that these are pilot research.

Author Response

Responses to reviewer 1:

Line 33. Delete the symbol "-" in "km-2"

R: we have corrected this as suggested

Line 116-117. Why did You collect soil samples in only one location at Eretan Kulon and two locations at Kertawinangun? It isn’t clear. Please explain it.

R:  Actually we collected one soil sample for each the three soil salinity levels. This was not to sample the village. We have changed the wording on Line 116-118 to clarify this. 

Line 124. Format equation as required in the file "agriculture-template.dot" in section 3.3. Formatting of Mathematical Components.

R: we have corrected this as suggested

Line 128. Use number of equation.

R: We have now removed this line because it is already described in the methods.

Section "3. Results and Discussion". It will be better if You divide this section into two separate sections "Results" and "Discussion".

R: We gave this careful consideration; however, due to complexity of the study we feel that it will be easier for readers to follow by having the results and discussion combined and then the conclusion to summarize and wrap up the findings.

Line 340-341. "This study, however, is a preliminary study and the results need to be verified with additional studies." Use a similar expression at the end of the section "1. Introduction". It is important to point out that these are pilot research.

R: We have added similar wording to Line 98 as suggested.

Reviewer 2 Report

The manuscript concerns imortant problem of agricultural use of highly saline soils. Soil salinity is a factor that significantly limits crop production, therefore negatively affectes food security. According to the presented results of study on highly saline soils a significant increase in rice yield can be achived by combination of proper fertilization and introduction of salt-tolerant variety. Improved rice production in this way can increase farmers' incom.

The recommended by authors fertilizer package include mineral  fertilizers, manure and gypsum. However, the chemical composition of manure is missing. Results are interesting but cognitive value of the manuscript could increase after adding content of available or/and total phosphorus and potassium in soils (table 1). The methodology for determination of organic matter, total N and CEC is also missing. 

Other specific comments:

References should be improved according to the requirements of the journal.

Line 107: longitude and latitude recording should be improved

In Table 1  the abberviation ESP need to be defined.

Line 122: the wording '' ...were measure in soil and water extracts 1:5'' is imprecise and should be ''... were measure in soil: water extracts of 1:5''

Line 144: the abberviation SP-36 need to be defined.

Line 285: I propose "..at P<0,05'' instead of ''...at the 5% levels" and other similar wording in all manuscript (e.g line 191)

Line 216: I propose full name test instead of abberviation DMRT

Author Response

Reviewer 2.

The recommended by authors fertilizer package include mineral  fertilizers, manure and gypsum. However, the chemical composition of manure is missing. Results are interesting but cognitive value of the manuscript could increase after adding content of available or/and total phosphorus and potassium in soils (table 1). The methodology for determination of organic matter, total N and CEC is also missing.

R:

The chemical composistion of manure is 1% N, 0.1%P2O5 and 0.1%K2O.  We have added to Line 151. 

Total Phosphorus and potassium, we have added in Table 1.

The methodology for determination of organic matter, total N and CEC have added on Line 123-124.

References should be improved according to the requirements of the journal.

references have been corrected as suggested.

Line 107: longitude and latitude recording should be improved

R:  we have corrected as suggested on Line 107-108.

In Table 1  the abberviation ESP need to be defined.

R: we have corrected as suggested as a table footnote

Line 122: the wording '' ...were measure in soil and water extracts 1:5'' is imprecise and should be ''... were measure in soil: water extracts of 1:5''

R: we have corrected this as suggested on Line 123.

Line 144: the abberviation SP-36 need to be defined.

R:  we have corrected as suggested on Line 153.

Line 285: I propose "..at P<0,05'' instead of ''...at the 5% levels" and other similar wording in all manuscript (e.g line 191)

R:  we have corrected this as suggested through out the manuscript.

Line 216: I propose full name test instead of abberviation DMRT

R: we have corrected this as suggested

Reviewer 3 Report

The article is well written, well-structured and complete. The results, however, concern a single year and this is a drawback, especially for field experiments where the weather conditions may induce a main impact on the results. For example, the differences in yields among saline and non-saline soils may differ significantly among dry and wet years that affect soil salinization. However, the authors do recognize this and they admit the preliminary importance of the results at the end. In that case, the authors should mention the year they conducted their research.

Author Response

Reviewer 3.

The authors should mention the year they conducted their research.

R:  The experiments were conducted during the dry season from August to November 2018.  This is described on line 113.